# A Survey of DEA Window Analysis Applications

Mohammed A. AlKhars *[ID], Ahmad H. Alnasser [ID] and Taqi AlFaraj

KFUPM Business School, King Fahd University of Petroleum and Minerals, Dhahran 31261, Saudi Arabia
* Correspondence: malkhars@kfupm.edu.sa

**Abstract:** This article aims to review, analyze, and classify the published research applications of the Data Envelopment Analysis (DEA) window analysis technique. The number of filtered articles included in the study is 109, retrieved from 79 journals in the web of science (WoS) database during the period 1996–2019. The papers are classified into 15 application areas: energy and environment, transportation, banking, tourism, manufacturing, healthcare, power, agriculture, education, finance, petroleum, sport, communication, water, and miscellaneous. Moreover, we present descriptive statistics related to the growth of publications over time, the journals publishing the articles, keyword terms used, length of articles, and authorship analysis (including institutional and country affiliations). To the best of the authors knowledge, this is the first survey reviewing the literature of the DEA window analysis applications in the 15 areas mentioned in the paper.

**Keywords:** DEA window analysis; efficiency; productivity; literature review; survey

## 1. Introduction

Data Envelopment Analysis (DEA) is a well-known mathematical technique, which is used to evaluate the relative efficiency of an individual organization, called a decision-making unit (DMU), in comparison with other organizations operating in a similar sector. Charnes et al. [1] published the first article using DEA to evaluate and compare the performance of a set of school districts participating in program follow through (PFT) and a set that did not. This paper was so influential that many variations of the DEA model have since been developed. The basic CCR model of DEA has been extended to several versions, and DEA window analysis is one of these several versions. Therefore, Gattoufi et al. [2] proposed a taxonomy to classify the DEA literature. They used four criteria: the data source used (D), the type of envelopment (E) invoked, the approach to analysis (A) used, and the nature (N) of the paper. Cook and Seiford [3] reviewed the major methodological development of DEA since its inception by Charnes et al. [1]. In a recent literature review by Emrouznejad and Yang (2018) [4], 10,300 general DEA journal articles were found in the period 1978–2016. Moreover, in another bibliographical study of DEA by Tavaresa (2002) [5], 3203 publication were identified, including journal papers, research papers, event papers, books, and dissertations. These bibliography studies have provided valuable information about DEA publications. However, it is more beneficial to conduct a literature review on a specific aspect of DEA. For example, Liu et al. [6] started with 4936 DEA papers retrieved from the ISI Web of Science (WOS) database published in the period from 1978 to August 2010. They classified the papers into two classifications—methodological (1802, 36.5%) and application-oriented (3134, 63.5%)—and focused on application-oriented papers, analyzing the development paths of the five major applications: banking, health care, agriculture and farms, transportation, and education. Similarly, Mardani et al. [7] reviewed 144 scholarly papers, published in 45 journals during the period 2006–2015, which used DEA in the energy efficiency field. They classified the papers into nine application fields: environmental efficiency, economic and eco-efficiency, energy efficiency issues, renewable and sustainable energy, water efficiency, energy performance, energy saving, integrated energy efficiency, and other application areas. Moreover, Soheilirad et al. [8] conducted

a literature review of 75 DEA articles published in 35 international journals and conferences in Supply Chain Management over the period 1996–2016. They classified the articles into eight application fields: sustainable supply chain, green supply chain, supply chain efficiency, supply chain performance, green and sustainable supplier selection, supplier selection, supplier performance, and other application areas. Finally, Mariz et al. [9] conducted a literature review of Dynamic Data Envelopment Analysis (DDEA) by reviewing one book and 79 articles published over the period 1996–2016 in the Scopus and Web of Science databases. They classified the articles into three categories: theoretical, practical, and theoretical and practical. Moreover, they analyzed the evolution of the DDEA literature over time.

In the current research, a literature review is conducted based on the use of the DEA window analysis approach. This method is important in two situations: for the first one, if the number of DMUs is small, then using DEA window analysis can increase the number of DMUs and consequently increase the discrimination power of the technique and make the results more robust. Second, DEA window analysis can help to track the performance of an organization over time and, therefore, allows better judgments across and within the windows compared to evaluating the performance during only one period [10,11]. This work surveys the application of DEA window analysis over 15 sectors. To the best of the authors' knowledge, this is the first time such a survey has taken place, which is expected to be appreciated by the scientific community.

The main purpose of this review paper is to provide an overview of the applications of the DEA window analysis technique. To achieve this purpose, the authors analyzed 109 articles published in 79 respected journals over the period 1996–2019, with the aim of answering the following questions: (1) What are the areas in which DEA window analysis has been applied? (2) What is the trend of using DEA window analysis? and (3) What are the affiliations and countries that have used DEA window analysis? We hope that this review can help researchers and scholars to obtain insight into the state-of-the-art in DEA window analysis research.

The remainder of this paper is organized as follows: Section 2 provides an overview of the DEA window analysis technique. Section 3 describes the research methodology and how the articles were retrieved, including the journals and publication trends over time. Section 4 presents an analysis of the review based on application areas, including the scope of the study, region of the study, number of windows, window width, and results obtained. Section 5 provides additional analyses of the keywords, length of papers, and authorship. Finally, Section 6 provides our conclusion, the limitations of the study, and suggestions for future research.

## 2. DEA Window Analysis

The first DEA model was introduced by Charnes, Cooper, and Rhodes and is known as the CCR model [1]. The mathematical formulation of the CCR model is given by:

$$Efficiency = Max \frac{\sum_r u_r y_{rk}}{\sum_i v_i x_{ik}},$$
$$\frac{\sum_r u_r y_{rj}}{\sum_i v_i x_{ij}} \leq 1, \; j = 1, \dots, n,$$
$$u_r, \; v_i \geq 0 \tag{1}$$

The above model considers a set of $n$ DMUs (DMU$_j$; $j = 1, \dots, n$) that consume $m$ inputs ($x_{ij}$; $i = 1, \dots, m$) to produce $s$ outputs ($y_{rj}$; $r = 1, \dots, s$), where $y_{rk}$ is the amount of the $r$th output from DMU$_k$, $u_r$ is the price weight given to the $r$th output, $x_{ik}$ is the amount of the $i$th input from DMU$_k$, and $v_i$ is the cost weight given to the $i$th input. The $k$th DMU is the one under consideration.

The CCR ratio model can be transformed into a mathematical linear model as follows:

$$Efficiency = Max \sum_r u_r y_{rk},$$
$$\sum_r u_r y_{rj} - \sum_i v_i x_{ij} \leq 0, \ j = 1, \ldots, n,$$
$$\sum_i v_i x_{ik} = 1,$$
$$u_r, \ v_i \geq 0$$

$$(2)$$

DEA window analysis is an extension of the CCR model, which evaluates the performance of DMUs over time. Charnes et al. [12] used DEA window analysis to evaluate the efficiency of maintenance units in the U.S. Air Force over a period of seven months. They used five windows, with each window spanning a period of three months. The use of DEA window analysis is useful in situations in which there is a small number of organizations or DMUs. In such cases, the use of DEA window analysis helps to effectively increase the number of DMUs. The relationship between the number of organizations, the width of the window, the number of windows, and the number of periods can be calculated by the following formula [10]:

$$w = k - p + 1,$$

$$Number \ of \ different \ organizations = n * p * w,$$

where:

$w$ = the number of windows,
$k$ = the number of periods,
$p$ = width of the windows,
$n$ = the number of organizations.

According to Asmild et al. [13], the selection of the window width should be as small as possible to reduce unfair comparisons over time but, at the same time, should be large enough to generate a sufficient sample size. As DEA window analysis evaluates performance over time, the time dimension should be added in the formulation. Continuing with the formulation presented in (2), let there be $n$ DMUs (DMU$_j$; $j = 1, \ldots, n$) that consume $m$ inputs (x$_{ij}$; $i = 1, \ldots, m$) to produce $s$ outputs (y$_{rj}$; $r = 1, \ldots, s$), observed in $T$ ($t = 1, \ldots, T$) periods. Let $DMU_k^t$ represent an observation $k$ in period $t$ having an input vector $X_k^t = \begin{bmatrix} x_k^{1t} \\ \vdots \\ x_k^{rt} \end{bmatrix}$ and an output vector $Y_k^t = \begin{bmatrix} y_k^{1t} \\ \vdots \\ y_k^{st} \end{bmatrix}$. Furthermore, consider a window that starts at time $l$ ($1 \leq l \leq T$) with a window width $w$ ($1 \leq w \leq T - l$). The matrices of the inputs and outputs are represented as follows:

$$X_{kw} = \begin{bmatrix} x_1^l & x_2^l & \cdots & x_n^l \\ x_1^{l+1} & x_2^{l+1} & \cdots & x_n^{l+1} \\ \vdots & \vdots & \ddots & \vdots \\ x_1^{l+w} & x_2^{l+w} & \cdots & x_n^{k+w} \end{bmatrix}, \ Y_{kw} = \begin{bmatrix} y_1^l & y_2^l & \cdots & y_n^l \\ y_1^{l+1} & y_2^{l+1} & \cdots & y_n^{l+1} \\ \vdots & \vdots & \ddots & \vdots \\ y_1^{l+w} & y_2^{l+w} & \cdots & y_n^{k+w} \end{bmatrix}$$

Substituting the inputs and outputs of $DMU_k^t$ into model (2), we can calculate the efficiency results of each DMU in the DEA window analysis.

## 3. Research Methodology

To conduct the research for classification of DEA window analysis, relevant observations were considered solely from articles within the Web of Science (WoS) database. Only the following four indices within the WoS were considered: The Science Citation Index Expanded, The Social Science Citation Index, The Arts & Humanities Citation Index, and The Emerging Sources Citation Index. The keywords used were "*window DEA*" and "*window data envelopment analysis*". The total number of articles found was 189. Five non-English articles were excluded. The remaining 184 articles were screened by titles,

abstracts and contents, from which 75 non-relevant articles were removed. Some of these articles used DEA but not the window analysis technique. Other papers used DEA to refer to another term, such as the *plasma DEA level.* After filtering, only 109 articles were found to qualify for the analysis. The process of identifying these articles was based on the recommendation of Preferred Reporting Items for Systematic Reviews and Meta-Analyses (PRISMA) Statement [14], as summarized in Figure 1. The authors tried their best to include all related articles, yet there is no guarantee that all relevant articles have been included.

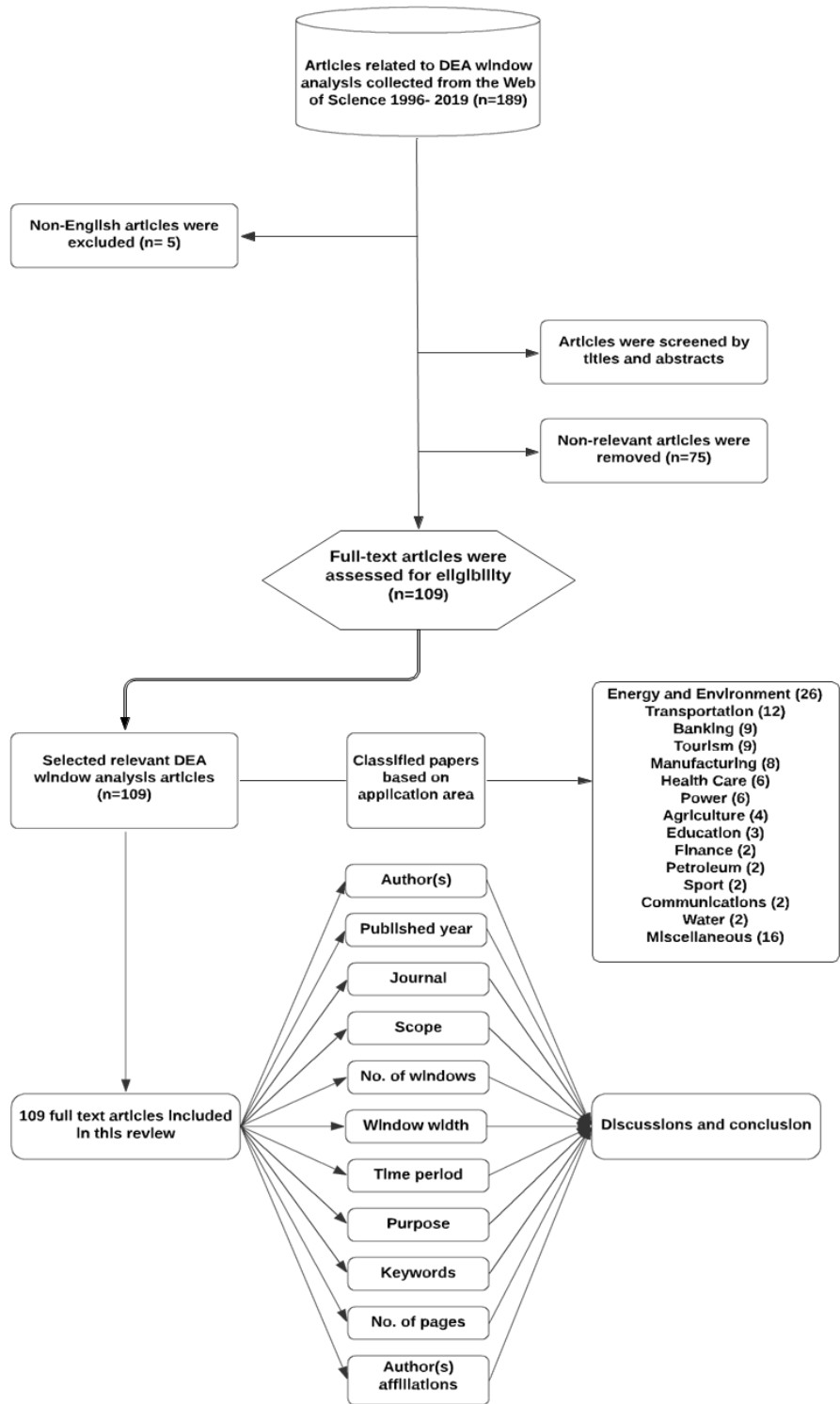

**Figure 1.** Analysis of articles related to Data Envelopment Analysis (DEA) window analysis.

The 109 articles were published in the period from 1996 to 2019. The number of publications during this period is presented in a histogram (Figure 2). The first article appeared in 1996, followed by a single or no article each year until 2007. The number of publications was three in 2008 and then increased over the years. The maximum number was in 2018, during which there were 27 publications.

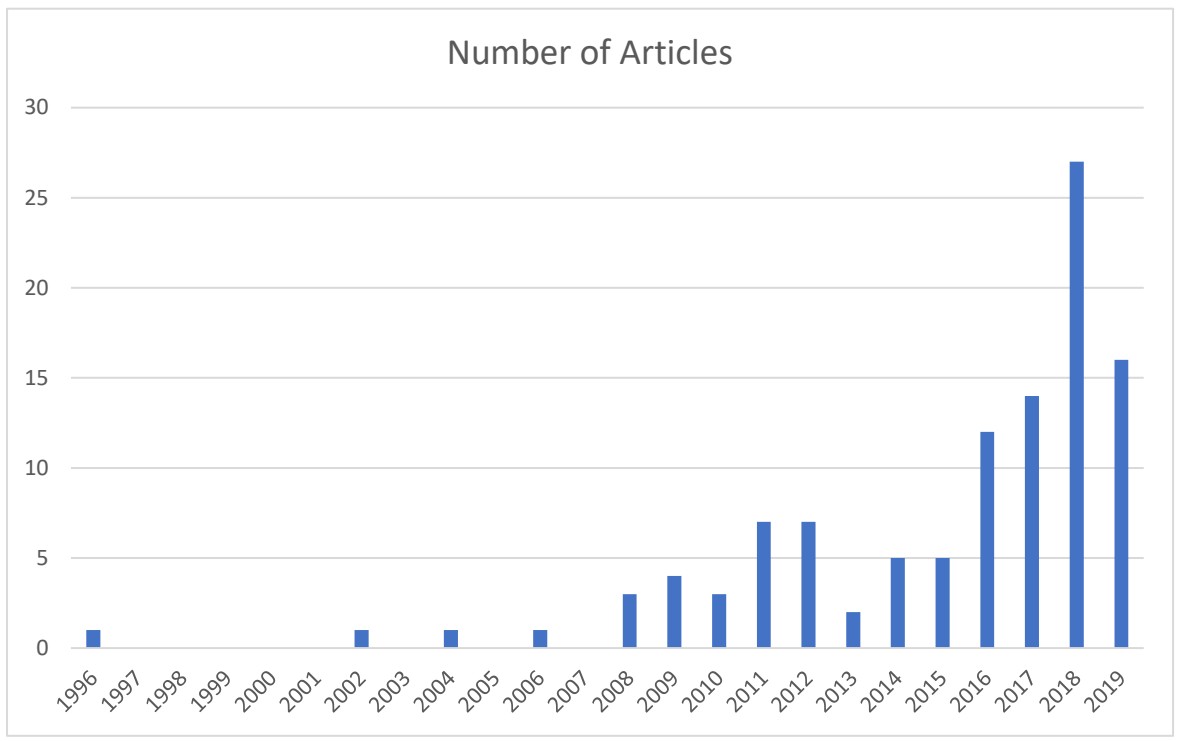

**Figure 2.** Number of publications over the years.

Moreover, the articles were published in 79 distinct journals. Table 1 shows the number of articles published in each journal. The highest number of publications was in the journal *Sustainability*, which published seven articles. Each of the journals *Economics*, *Energy Policy*, *Expert Systems with Applications*, and *Journal of Clean Production* published four articles.

**Table 1.** List of journals publishing DEA window analysis articles.

| No. | Journal Name | Frequency |
|---|---|---|
| 1 | *Sustainability* | 7 |
| 2 | *Applied Economics* | 4 |
| 3 | *Energy Policy* | 4 |
| 4 | *Expert System with Applications* | 4 |
| 5 | *Journal of Cleaner Production* | 4 |
| 6 | *Croatian Operational Research Review* | 3 |
| 7 | *Journal of Productivity Analysis* | 3 |
| 8 | *Benchmarking: An International Journal* | 2 |
| 9 | *Ecological Economics* | 2 |
| 10 | *Energy Efficiency* | 2 |
| 11 | *International Journal of Production Economics* | 2 |
| 12 | *Journal of Policy Modeling* | 2 |
| 13 | *Renewable and Sustainable Energy Reviews* | 2 |
| 14 | *Tertiary Education and Management* | 2 |
| 15 | *Tourism Economics* | 2 |
| 16 | *African Journal of Agricultural Research* | 1 |
| 17 | *African Journal of Business Management* | 1 |

**Table 1.** *Cont.*

| No. | Journal Name | Frequency |
| --- | --- | --- |
| 18 | *Applied Economics Letters* | 1 |
| 19 | *Asian Journal of Shipping and Logistics* | 1 |
| 20 | *BMC Health Services Research* | 1 |
| 21 | *Brazilian Journal of Operations & Production Management* | 1 |
| 22 | *Bulgarian Chemical Communications* | 1 |
| 23 | *Central European Journal of Operations Research* | 1 |
| 24 | *Chinese Journal of Urban and Environmental Studies* | 1 |
| 25 | *DRVNA INDUSTRIJA* | 1 |
| 26 | *Ecological Indicators* | 1 |
| 27 | *Economic Computation and Economic Cybernetics Studies and Research* | 1 |
| 28 | *Economic Modelling* | 1 |
| 29 | *Economic Research-Ekonomska Istraživanja* | 1 |
| 30 | *Ekonomicky Casopis* | 1 |
| 31 | *Energy Economics* | 1 |
| 32 | *Environmental Progress & Sustainable Energy* | 1 |
| 33 | *Environmental Science & Policy* | 1 |
| 34 | *Environmental Science and Pollution Research* | 1 |
| 35 | *European Journal of Operational Research* | 1 |
| 36 | *European Journal of Operations Research* | 1 |
| 37 | *Geosystem Engineering* | 1 |
| 38 | *Global Economic Review* | 1 |
| 39 | *Health Economics Review* | 1 |
| 40 | *Health Policy and Planning* | 1 |
| 41 | *International Journal of Innovation and Sustainable Development* | 1 |
| 42 | *International Journal of Life Cycle Assessment* | 1 |
| 43 | *International Journal of Logistics Research and Applications* | 1 |
| 44 | *International Journal of Performance Analysis in Sport* | 1 |
| 45 | *International Journal of Productivity and Performance Management* | 1 |
| 46 | *International Journal of Tourism Research* | 1 |
| 47 | *Inzinerine Ekonomika (Engineering Economics)* | 1 |
| 48 | *Jordan Journal of Mechanical and Industrial Engineering* | 1 |
| 49 | *Journal of Business Research* | 1 |
| 50 | *Journal of Comparative Effectiveness Research* | 1 |
| 51 | *Journal of Environmental Management* | 1 |
| 52 | *Journal of Global Operations and Strategic Sourcing* | 1 |
| 53 | *Journal of Hospitality Marketing & Management* | 1 |
| 54 | *Journal of Industrial Ecology* | 1 |
| 55 | *Journal of Operations Management* | 1 |
| 56 | *Journal of Scientific & Industrial Research* | 1 |
| 57 | *Journal of the Operational Research Society* | 1 |
| 58 | *Journal of the Operations Research Society of Japan* | 1 |
| 59 | *Marine Policy* | 1 |
| 60 | *Mathematical and Computer Modelling* | 1 |
| 61 | *Neural Computing and Applications* | 1 |
| 62 | *OR Spectrum* | 1 |
| 63 | *Plos One* | 1 |
| 64 | *Promet-Traffic & Transportation* | 1 |
| 65 | *Renewable Energy* | 1 |
| 66 | *Resources Policy* | 1 |
| 67 | *Resources, Conservation and Recycling* | 1 |
| 68 | *Science and Public Policy* | 1 |
| 69 | *Scientometrics* | 1 |
| 70 | *Sigma Journal of Engineering and Natural Sciences* | 1 |
| 71 | *Social Indicators Research* | 1 |
| 72 | *Sosyoekonomi* | 1 |
| 73 | *Symmetry* | 1 |

**Table 1.** *Cont.*

| No. | Journal Name | Frequency |
|---|---|---|
| 74 | *Technology Analysis & Strategic Management* | 1 |
| 75 | *Telecommunications Policy* | 1 |
| 76 | *The Asian Journal of Shipping and Logistics* | 1 |
| 77 | *Tourism, Turizam: međunarodni znanstveno-stručni časopis* | 1 |
| 78 | *Transportation Planning and Technology* | 1 |
| 79 | *ZBORNIK RADOVA EKONOMSKOG FAKULTETA U RIJECI-PROCEEDINGS OF RIJEKA FACULTY OF ECONOMICS* | 1 |
| | Total | 109 |

## 4. Classification of DEA Window Analysis Applications

In line with Liu et al. [6], who surveyed the DEA applications and utilized 26 application areas, we classified the reviewed articles into the following 15 application areas: energy and environment, transportation, banking, tourism, manufacturing, healthcare, power, agriculture, education, finance, petroleum, sport, communication, water, and miscellaneous. Table 2 presents the number and percentage of articles in each application area. A total of 26 articles was published in the energy and environment area, representing about 24% of articles, while 12 articles were published in the transportation area, representing around 11%. The fewest number of articles was published in the areas of finance, petroleum, sport, communication, and water, each with two articles. Articles that did not fit in the first 14 application areas were classified as miscellaneous. Tables 3–17 summarize the articles in each application area. The authors tried their best to provide as much accurate information as possible; however, there may be some unintended mistakes.

**Table 2.** Number and percentage of articles in each application area.

| No. | Application Area | Frequency | % |
|---|---|---|---|
| 1 | Energy & Environment | 26 | 24% |
| 2 | Transportation | 12 | 11% |
| 3 | Banking | 9 | 8% |
| 4 | Tourism | 9 | 8% |
| 5 | Manufacturing | 8 | 7% |
| 6 | Healthcare | 6 | 6% |
| 7 | Power | 6 | 6% |
| 8 | Agriculture | 4 | 4% |
| 9 | Education | 3 | 3% |
| 10 | Finance | 2 | 2% |
| 11 | Petroleum | 2 | 2% |
| 12 | Sport | 2 | 2% |
| 13 | Communication | 2 | 2% |
| 14 | Water | 2 | 2% |
| 15 | Miscellaneous | 16 | 15% |
| | Total | 109 | 100% |

**Table 3.** Articles classified under the energy and environment category.

| Authors and Year | Country | Scope | No. of Windows | Window Width | Time Period | Purpose |
|---|---|---|---|---|---|---|
| Halkos and Tzeremes (2009) [11] | Multiple countries | 17 OECD Countries | 22 | 3 | 1980–2002 | To study the existence of the Kuznets relationship between the environmental efficiency and national income of countries. |
| Zhang et al. (2011) [15] | Multiple countries | 23 developing countries | 24 | 3 | 1980–2005 | To study energy efficiency in 23 developing countries from 1980 to 2005. |
| Vlahinić-Dizdarević and Šegota (2012) [16] | Multiple countries | 26 EU countries | 9 | 3 | 2000–2010 | To study the efficiency changes of energy in EU countries in the period 2000–2010. |

**Table 3.** *Cont.*

| Authors and Year | Country | Scope | No. of Windows | Window Width | Time Period | Purpose |
|---|---|---|---|---|---|---|
| Wang et al. (2012) [17] | China | 30 regions in China | 8 | 3 | 2000–2009 | To assess the total-factor energy and emissions performance of 30 regions in China. |
| Wang et al. (2013) [18] | China | 29 Administrative Regions of China | 7 | 3 | 2000–2008 | To investigate the total-factor energy and environmental efficiency in 29 regions in China. |
| Wu et al. (2014) [19] | China | 30 regions in China | 4 | 3 | 2005–2010 | To assess the circular economy efficiency of 30 regions in China from 2005 to 2010. |
| Camioto et al. (2014) [20] | Brazil | seven sectors | 7 | 8 | 1996–2009 | To assess the efficiency of industrial sectors in Brazil during the period 1996–2009. |
| Camioto et al. (2016) [21] | Multiple countries | 12 countries | 9 | 10 | 1993–2010 | To examine the total-factor energy efficiency in BRICS countries (Brazil, Russia, India, China, and South Africa) and G7 countries (Canada, France, Germany, Italy, Japan, the United Kingdom, and the United States) while considering the total-factor structure. |
| Yang et al. (2016) [22] | Taiwan | Taiwan's 22 Administrative Regions | 5 | 2 | 2006–2011 | To measure the urban sustainability and the aggregate urban input–output efficiency of 22 administrative regions in Taiwan. |
| Halkos et al. (2016) [23] | Multiple countries | 20 countries | 18 | 5 | 1990–2011 | To evaluate the sustainability efficiency of 20 advanced-economy countries over the period 1990–2011. |
| Al-Refaie et al. (2016) [24] | Jordan | Jordan Industrial Sector | 11 | 5 | 1999–2013 | To evaluate the growth of the energy efficiency and productivity in the industrial sector from 1999 to 2013. |
| Lv et al. (2017) [25] | China | 30 regions | 8 | 3 | 2001–2010 | To assess the energy efficiency from 2001 to 2010 in China. |
| Camioto et al. (2017) [26] | Brazil | seven industrial sectors | 8 | 8 | 1996–2010 | To assess the efficiency of industrial sectors in Brazil in the period 1996–2009. |
| Sueyoshi et al. (2017) [27] | China | 30 provinces | 10 | 3 | 2003–2014 | To evaluate the energy and environmental efficiency in 30 provinces of China from 2003 to 2014. |
| Rahbari et al. (2018) [28] | Iran | 24 samples | 4 | 3 | 2009–2014 | To measure the efficiency of the Khuzestan steel company treatment plant. |
| Li et al. (2018) [29] | China | 30 Regional Industrial Systems in China | 5 | 3 | 2004–2010 | To measure the environmental efficiency of industrial systems in 30 regions in China. |
| Lorenzo-Toja et al. (2018) [30] | Spain | 47 wastewater treatment plants | 4 | 1 | 2009–2012 | To evaluate the environmental sustainability of wastewater treatment plants. |
| Fu et al. (2018) [31] | China | 30 regions in China | 9 | 2 | 2006–2015 | To assess the efficiency of the industrial green transformation in 30 regions in China in the period 2006–2015. |
| Zhang et al. (2018) [32] | China | 30 provinces | 8 | 3 | 2007–2014 | To assess the performance of 30 Chinese provinces in the period 2007–2014. |
| Zhang et al. (2018) [33] | Multiple countries | 16 countries | 24 | 3 | 1990–2015 | To assess the total factor energy efficiency and carbon emissions performance of top countries participating in CDM projects from 1990 to 2015. |
| Li et al. (2018) [34] | China | 25 cities | 9 | 3 | 2000–2010 | To examine the consequence of urbanization on $CO_2$ emissions efficiency. |
| Camioto et al. (2018) [35] | Multiple countries | 15 Latin American countries | 12 | 12 | 1991–2013 | To evaluate the renewable energy sources and energy efficiency of 15 Latin American countries. |
| Wang et al. (2018) [36] | Canada | four Canadian wastewater treatment plants | 10, 6, 1 | 1, 5, 10 | 2007–2016 | To evaluate the efficiency of four Canadian WWTPs during the period 2007–2016. |

**Table 3.** *Cont.*

| Authors and Year | Country | Scope | No. of Windows | Window Width | Time Period | Purpose |
|---|---|---|---|---|---|---|
| Kupeli et al. (2019) [37] | Multiple countries | 35 countries in the OECD | 5 | 2 | 2010–2015 | To assess the renewable energy performances of 35 OECD countries. |
| Wang et al. (2019) [38] | China | China's 30 provinces | 12 | 3 | 2003–2016 | To evaluate the carbon emissions efficiency of 30 provinces in China from 2003 to 2016. |
| Yu (2019) [39] | Taiwan | 19 Administrative Regions of Taiwan | 4 | 3 | 2011–2016 | To evaluate the sustainable development efficiency across 19 administrative regions of Taiwan during the period 2011–2016. |

**Table 4.** Articles classified under the transportation category.

| Authors and Year | Country | Scope | No. of Windows | Window Width | Time Period | Purpose |
|---|---|---|---|---|---|---|
| Pjevčević et al. (2012) [40] | Serbia | five ports | 5 | 4 | 2001–2008 | To analyze the efficiency of five ports in Serbia. |
| Yang (2012) [41] | Taiwan | four ports | 5 | 3 | 2001–2007 | To evaluate the productivity changes in the port industry in Taiwan from 2003 to 2007. |
| Min et al. (2015) [42] | USA | 24 urban mass transit agencies | 3 | 1 | 2009–2011 | To assess the operational efficiency of the urban mass transit agencies in the U.S. |
| Liu et al. (2016) [43] | China | 30 provinces in China | 13 | 3 | 1998–2012 | To assess the energy and environment efficiency of the road and railway sectors in 30 regions in China. |
| Song et al. (2016) [44] | China | 30 provinces in China | 2 | 1 | 2011–2012 | To measure the environmental regional efficiency of highway transportation systems in China. |
| Rabar et al. (2017) [45] | Croatia | seven Croatian airports | 1 | 6 | 2009–2014 | To investigate the efficiency of seven Croatian airports from 2004 to 2008. |
| Park et al. (2018) [46] | South Korean | 10 Regional Offices of Oceans and Fisheries (ROOFs) | 8 | 3 | 2007–2016 | To assess the operational efficiency of the South Korean coastal ferry industry. |
| Chen et al. (2018) [47] | China | 15 cities | 3 | 3 | 2009–2013 | To assess the transportation energy efficiency of 15 cities in the Yangtze River Delta during the period 2009–2013. |
| Wang et al. (2019) [48] | Multiple countries | 16 Asia airline companies | 3 | 3 | 2012–2016 | To assess the performance of 16 major Asian airline companies. |
| Yang et al. (2019) [49] | China | 14 cities of Hunan province | 3 | 3 | 2012–2016 | To assess the urban road transport and land-use efficiency in 14 cities of Hunan province, central China, during the period 2012–2016. |
| George and Tumma (2019) [50] | India | 13 major seaports of India | 3 | 1 | 2014–2016 | To evaluate the operational and financial performances of 13 major Indian seaports. |
| Zarbi et al. (2019) [51] | Iran | 5 ports | 7 | 4 | 2012–2018 | To assess the performance and relative efficiency of 5 ports in Iran. |

**Table 5.** Articles classified under the banking category.

| Authors and Year | Country | Scope | No. of Windows | Window Width | Time Period | Purpose |
|---|---|---|---|---|---|---|
| Hartman and Storbeck (1996) [52] | Sweden | 12 Swedish banks | 3 | 3 | 1984–1992 | To assess the efficiency of loan operations in 12 Swedish banks from 1984 to 1992. |
| Asmild et al. (2004) [13] | Canada | Five large participant banks | 16 | 5 | 1981–2000 | To assess the performance of the banking industry in Canada. |
| Nguyen et al. (2014) [53] | Vietnam | Banking sector | 15 | 3 | 1995–2011 | To evaluate the efficiency of the Vietnamese banking sector from 1995 to 2011. |
| Shawtari et al. (2015) [54] | Yemen | 16 banks | 14 | 3 | 1996–2011 | To evaluate the efficiency of the banking industry in Yemen. |

**Table 5.** *Cont.*

| Authors and Year | Country | Scope | No. of Windows | Window Width | Time Period | Purpose |
|---|---|---|---|---|---|---|
| Tuškan and Stojanović (2016) [55] | Multiple countries | 28 European banking systems | 5 | 1 | 2008–2012 | To assess the efficiency of the banking industry of 28 European banking systems from 2008 to 2012. |
| Cvetkoska and Savić (2017) [56] | Republic of Macedonia | Eight branches | 2 | 2 | 2009–2011 | To evaluate the efficiency of the branches of Komercijalna Banka AD Skopje during the period 2009–2011. |
| Degl'Innocenti et al. (2017) [57] | 9 EU members | 116 banks | 10 | 3 | 2004–2015 | To study the efficiency of 116 banks of nine new EU members in Central and Eastern European (CEE) countries from 2004 to 2015. |
| Phan et al. (2018) [58] | Hong Kong | 41 financial institutions | 9 | 3 | 2004–2014 | To evaluate the cost efficiency of the Banking sector in Hong Kong from 2004 to 2014. |
| Shawtari et al. (2018) [59] | Taiwan | Taiwan's 22 administrative regions | 5 | 2 | 2006–2011 | To evaluate the urban sustainability and the aggregate urban input–output efficiency of 22 administrative regions in Taiwan. |

**Table 6.** Articles classified under the tourism category.

| Authors and Year | Country | Scope | No. of Windows | Window Width | Time Period | Purpose |
|---|---|---|---|---|---|---|
| Yang and Lu (2006) [60] | Taiwan | 46 international tourist hotels (ITHs) in Taiwan | 4 | 3 | 1997–2002 | To evaluate the operational performance of 46 Taiwanese international tourist hotels (ITHs) from 1997 to 2002. |
| Liu (2008) [61] | UK | 13 theme parks | 8 | 3 | 1997–2006 | To evaluate the financial performance of 13 theme parks in the UK. |
| Pulina et al. (2010) [62] | Italy | 21 regions in Italy | 2 | 2 | 2000–2002 | To evaluate the efficiency of hotels across all 20 Italian regions. |
| Huang et al. (2012) [63] | China | 31 geographical regions | 4 | 3 | 2001–2006 | To investigate the technical efficiency of the hotel industry at the regional level. |
| Detotto et al. (2014) [64] | Italy | 21 regions | 3 | 3 | 2000–2004 | To examine the productivity of the hospitality sector at the regional level in Italy. |
| Ohe and Peypoch (2016) [65] | Japan | 9 regions | 7 | 2 | 2005–2012 | To assess the efficiency of Japanese ryokans from 2005 to 2012. |
| Xu and Chi (2017) [66] | USA | Six types of hotel | 6 | 3 | 2007–2014 | To study the operating efficiency of U.S. hotels during the period 2007–2014. |
| Cuccia et al. (2017) [67] | Italy | 21 Italian regions | 15 | 3 | 1995–2010 | To examine the effect of United Nations Educational Scientific and Cultural Organization (UNESCO) sites on the enhancement of tourism destinations (TDs) performance in Italy during the period 1995–2010. |
| Škrinjarić (2018) [68] | Croatia | 21 Croatian counties | 4 | 2 | 2011–2015 | To assess the efficiency of the tourism industry of 21 Croatian counties from 2011 to 2015. |

**Table 7.** Articles classified under the manufacturing category.

| Authors and Year | Country | Scope | No. of Windows | Window Width | Time Period | Purpose |
|---|---|---|---|---|---|---|
| Chung et al. (2008) [69] | Taiwan | Seven mixes | 5 | 3 | Unspecified | To assess the efficiency of product family mixes in a wafer fab. |
| Lee and Pai (2011) [70] | Taiwan, Korea, and Japan | 10 TF–LCD firms | 4 | 3 | 2002–2007 | To evaluate the operational efficiency of global TFT–LCD firms. |
| Hemmasi et al. (2011) [71] | Iran | 10 firms in the Iranian wood panels industry | 3 | 3 | 2002–2006 | To assess the performance of 10 firms in the Iranian wood panels industry from 2002 to 2006. |
| Lin et al. (2018) [72] | China | 28 manufacturing industries | 5 | 5 | 2006–2014 | To assess the efficiency of green technology innovation in 28 Chinese manufacturing industries from 2006 to 2014. |

**Table 7.** *Cont.*

| Authors and Year | Country | Scope | No. of Windows | Window Width | Time Period | Purpose |
|---|---|---|---|---|---|---|
| Lee et al. (2018) [73] | China, Korea, and Japan | 10 firms | 6 | 3 | 2002–2009 | To evaluate the operational performance of 10 major TFT–LCD (thin film transistor–liquid crystal display) manufacturers in China, Korea, and Japan. |
| Kropivšek and Grošelj (2019) [74] | Slovenia | 2 sub-sectors | 6 | 5 | 2007–2016 | To investigate the performance of the Slovenian wood industry. |
| Al-Refaie et al. (2019) [75] | Jordan | three blister packing lines (BL1, BL2, and BL3) | 14 | 6 | January 2013–December 2014 | To assess the efficiency of blistering lines on a monthly basis from January 2013 to December 2014. |
| Apan et al. (2019) [76] | Turkey | 19 firms | 8 | 3 | 2008–2017 | To examine the financial activities of 19 firms in the textile sector being traded on Borsa Istanbul (BIST) for the period 2008–2017. |

**Table 8.** Articles classified under the healthcare category.

| Authors and Year | Country | Scope | No. of Windows | Window Width | Time Period | Purpose |
|---|---|---|---|---|---|---|
| Jia and Yuan (2017) [77] | China | 5 hospitals | 5 | 3 | Unspecified | To evaluate and compare the operational efficiencies of different hospitals before and after establishing their branched hospitals. |
| Flokou et al. (2017) [78] | Greece | 107 Greek NHS hospitals | 4 | 2 | 2009–2013 | To evaluate the efficiency of 107 Greek NHS hospitals from 2009 to 2013. |
| Stefko et al. (2018) [79] | Slovakia | 8 regions | 5 | 4 | 2008–2015 | To assess the efficiency of healthcare facilities in eight regions in Slovakia from 2008 to 2015. |
| Serván-Mori et al. (2018) [80] | Mexico | 233 health jurisdictions | Unspecified | Unspecified | 2008–2015 | To measure the level of the technical efficiency of the primary care units in Mexico. |
| Kocisova et al. (2019) [81] | Slovakia | 8 Slovak regions | 8 | 1 | 2008–2015 | To assess the technical efficiency of the healthcare facilities in eight regions in Slovakia from 2008 to 2015. |
| Fuentes et al. (2019) [82] | Spain | Nine acute general hospitals | 1 | 3 | 2012–2014 | To assess the efficiency of public acute hospitals located in the Murcia region in Spain. |

**Table 9.** Articles classified under the power category.

| Authors and Year | Country | Scope | No. of Windows | Window Width | Time Period | Purpose |
|---|---|---|---|---|---|---|
| Sözen et al. (2012) [83] | Turkey | 10 hydro-power plants (HPPs) | 2 | 2 | 2007–2009 | To evaluate the performance of ten hydro-power plants (HPP) in Turkey. |
| Bono and Giacomarra (2016) [84] | Multiple countries | 11 EU countries | 14 | 5 | 1996–2010 | To measure the technical efficiency performances of the photovoltaic sector in EU countries from 1996 to 2010. |
| Song et al. (2017) [85] | China | 28 coal-fired power generation sectors | 3 | 3 | 2006–2010 | To assess the performance of the power generation industry in China. |
| Barabutu and Lee (2018) [86] | South Africa | 12 state-owned electric companies | 9 | 4 | 2004–2015 | To evaluate the efficiency of twelve (12) state-owned electric companies operating in the Southern African Power Pool (SAPP) from 2004 to 2015. |
| Halkos and Polemis (2018) [87] | USA | 50 states in the U.S. | 11 | 3 | 2000–2012 | To evaluate the efficiency of the power generation sector in 50 states in the U.S. |
| Sun et al. (2018) [88] | China | 30 provinces in China | 9 | 3 | 2005–2015 | To evaluate the efficiency of the fossil fuel power plants in China. |

**Table 10.** Articles classified under the agriculture category.

| Authors and Year | Country | Scope | No. of Windows | Window Width | Time Period | Purpose |
|---|---|---|---|---|---|---|
| Masuda (2016) [89] | Japan | 2 fields | 9 | 9 | 1995–2011 | To evaluate the eco-efficiency of wheat production in Japan. |
| Vlontzos and Pardalos (2017) [90] | Multiple countries | 25 EU members | 5 | 3 | 2006–2012 | To evaluate GHG emissions efficiency in 25 EU countries. |
| Masuda (2018) [91] | Japan | 9 scales of rice farms | 4 | 4 | 2005–2011 | To study the consequence of increasing the scale of rice farming on the energy efficiency of intensive rice production in Japan. |
| Masuda (2019) [92] | Japan | 9 farm sizes | 4 | 4 | 2005–2011 | To study if expanding the scale of rice farming leads to improving the eco-efficiency of intensive rice production in Japan. |

**Table 11.** Articles classified under the education category.

| Authors and Year | Country | Scope | No. of Windows | Window Width | Time Period | Purpose |
|---|---|---|---|---|---|---|
| Lee et al. (2012) [93] | Republic of Korea | 23 public research institutions (PRIs) | 1 | 11 | 2000–2010 | To examine the effect of co-operating forms on the R&D performance of public research institutions (PRIs) in Korean science and engineering fields. |
| Guccio et al. (2017) [94] | Italy | 54 Italian public universities | 9 | 3 | 2000–2010 | To evaluate the efficiency of public universities in Italy from 2000 to 2010. |
| Moreno et al. (2019) [95] | Spain | 47 universities | 4 | 4 | 2009–2015 | To evaluate the efficiency of 47 public universities in Spain during the period 2008/9–2014/15. |

**Table 12.** Articles classified under the finance category.

| Authors and Year | Country | Scope | No. of Windows | Window Width | Time Period | Purpose |
|---|---|---|---|---|---|---|
| Sun (2011) [96] | Taiwan | 13 financial holdings companies in Taiwan | 5 | 3 | 2003–2009 | To examine the current evaluation system of 13 financial holdings companies in Taiwan. |
| Zhang and Chen (2018) [97] | Multiple countries | 11 energy investment schemes | 38 | 3 | Q12006–Q42015 | To assess the performance of 11 energy investment schemes. |

**Table 13.** Articles classified under the petroleum category.

| Authors and Year | Country | Scope | No. of Windows | Window Width | Time Period | Purpose |
|---|---|---|---|---|---|---|
| Ross and Droge (2001) [98] | Multiple countries | 102 distribution centers (DCs) | 3 | 2 | 1993–1996 | To evaluate the productivity of 102 distribution centers (DCs) in the period 1993–1996. |
| Sueyoshi and Wang (2018) [99] | USA | 30 companies | 4 | 2 | 2012–2016 | To evaluate the performance of 30 companies in the petroleum industry in the United States (U.S.) |

**Table 14.** Articles classified under the sport category.

| Authors and Year | Country | Scope | No. of Windows | Window Width | Time Period | Purpose |
|---|---|---|---|---|---|---|
| Lin et al. (2016) [100] | China | 4 teams | 6 | 3 | 2007–2014 | To evaluate the offense efficiency, defense efficiency, and integrated efficiency of four teams in the CPBL during the period 2007–2014. |
| García-Cebrián et al. (2018) [101] | Multiple countries | 32 teams | 7 | 3 | 2004–2012 | To study the efficiency of teams playing in the UEFA Champions League during the seasons 2004–2012. |

**Table 15.** Articles classified under the communication category.

| Authors and Year | Country | Scope | No. of Windows | Window Width | Time Period | Purpose |
|---|---|---|---|---|---|---|
| Resende and Tupper (2009) [102] | Brazil | 39 Brazilian companies | 1 | 14 | February 2000–May 2003 | To evaluate the quality efficiency of Brazilian mobile companies from 2000 to 2003. |
| Yang and Chang (2009) [10] | Taiwan | 3 leading firms | 13 | 8 | Q12001–Q42005 | To evaluate the efficiency of three telecommunication firms from 2001 to 2005. |

**Table 16.** Articles classified under the water category.

| Authors and Year | Country | Scope | No. of Windows | Window Width | Time Period | Purpose |
|---|---|---|---|---|---|---|
| Luo et al. (2018) [103] | China | 12 western Chinese provinces | 9 | 3 | 2005–2015 | To measure the water use efficiency in 12 western provinces in China in the period 2005–2015. |
| Wang (2018) [104] | China | 31 provinces | 6 | 3 | 2005–2012 | To study water resources efficiency in China from 2005 to 2012. |

**Table 17.** Articles classified under the miscellaneous category.

| Authors and Year | Country | Scope | No. of Windows | Window Width | Time Period | Purpose |
|---|---|---|---|---|---|---|
| Halkos and Tzeremes (2008) [105] | Multiple countries | 16 OECD countries | 3 | 3 | 1996–2000 | To measure the "trade efficiency" in 16 OECD countries in order to determine the factors influencing the relationship between development and trade growth. |
| Halkos and Tzeremes (2009) [106] | Multiple countries | 25 EU members | 9 | 3 | 1995–2005 | To assess the economic efficiency of growth policies of the 25 EU countries. |
| Halkos and Tzeremes (2010) [107] | Multiple countries | 79 countries | 6 | 3 | 2000–2006 | To examine the impact of corruption on the economic efficiency of countries. |
| Cullinane and Wang (2010) [108] | Multiple countries | 25 leading container ports | 6 | 3 | 1992–1999 | To examine the efficiency of 25 ports from 1992 to 1999. |
| Sun (2011) [109] | Taiwan | 6 industries in Taiwan | 5 | 3 | 2000–2006 | To investigate the growth of efficiency and productivity of six industries in Taiwan Hsin Chu Industrial Science Park from 2000 to 2006. |
| Halkos and Tzeremes (2011) [110] | Multiple countries | 42 countries | 19 | 3 | 1986–2006 | To examine the relationship between economic efficiency and oil consumption in 42 countries from 1986 to 2006. |
| Chien et al. (2011) [111] | Multiple countries | 10 ASEAN countries | 2 | 2 | 2001–2003 | To assess technology efficiency and effectiveness in 10 ASEAN countries. |
| Vázquez-Rowe and Tyedmers (2013) [112] | USA | 4 ports | 34 | 1 | 2001 | To monitor, calculate, and quantify the inefficiency resulting from the "skipper effect". |
| Škare and Rabar (2014) [113] | Croatia | 21 counties | 3 | 1 | 2005–2007 | To evaluate regional efficiency in Croatia from 2005 to 2007. |
| Rabar (2015) [114] | Croatia | 5 Croatian shipyards | 6 | 1 | 2007–2012 | To evaluate the relative efficiency of five shipyards in Croatia. |
| Santana et al. (2015) [115] | Multiple countries | 12 countries | 5 | 5 | 2000–2008 | To examine the efficiency of BRICS and G7 countries to transform national innovative capacity into economic, environmental, and social development in the period 2000–2008. |
| Hunjet et al. (2015) [116] | Croatia | 12 towns | 4 | 3 | 2004–2009 | To evaluate the efficiency of 12 towns in Croatia. |
| Al-Refaie et al. (2016) [117] | Unspecified | 5 blowing machines | 7 | 6 | February 2014–July 2014 | To evaluate the efficiency of five blowing machines in the plastics industry in both day and night shifts from February 2014 to June 2014. |

**Table 17.** *Cont.*

| Authors and Year | Country | Scope | No. of Windows | Window Width | Time Period | Purpose |
|---|---|---|---|---|---|---|
| Skare and Rabar (2017) [118] | China | 30 OECD countries | 10 | 1 | 2002–2011 | To examine the socio-economic efficiency of thirty OECD countries. |
| Liu et al. (2019) [119] | Iran | 6 fields of study | 5 | 3 | 2002–2012 | To assess the performance of research projects in six main fields of study handled by the Ministry of Science and Technology (MOST) in Taiwan during the period 2002–2012. |
| Lin et al. (2019) [120] | China | 7 types of Chinese industrial enterprises | 5 | 6 | 2006–2015 | To assess the efficiency of the technological innovation of seven types of industrial enterprises in China from 2006 to 2015. |

## 5. Statistics on DEA Window Analysis Publications

This section provides additional descriptive statistics about the reviewed DEA window analysis articles according to the following: keywords, length of articles, number of authors per article, and author affiliations.

### 5.1. Statistics Based on Keywords

Table 18 provides a list of the top five keyword terms used in the reviewed articles. The most-used keyword term was "DEA window analysis" and its variants. These keyword terms appeared 69 times in the surveyed articles. The next most-used keyword term was "DEA" and its variants, which appeared 63 times. The keyword terms "efficiency", "efficiency evaluation", and efficiency measurement" appeared 19 times. "$CO_2$ emission" and its variants appeared seven times. Finally, "energy efficiency" appeared five times. The last two keywords ($CO_2$ emission and energy efficiency) were consistent with the previous analysis, as the articles in the energy and environment category were ranked first, in terms of DEA window analysis publications.

**Table 18.** Top five most-used keyword terms.

| No. | Keyword | Frequency |
|---|---|---|
| 1 | data envelopment analysis window analysis; data envelopment window analysis; DEA window; DEA window analysis; DEA–window, window analysis; window data envelopment analysis; window DEA | 69 |
| 2 | data envelope analysis (DEA); data envelopment analysis; DEA, DEA analysis | 63 |
| 3 | efficiency; efficiency evaluation; efficiency measurement | 19 |
| 4 | carbon dioxide emissions; $CO_2$ emission; $CO_2$ emissions efficiency; emissions efficiency | 7 |
| 5 | energy efficiency | 5 |

### 5.2. Statistics Based on Number of Pages (Size)

More than 1500 pages were published related to DEA window analysis in the 109 reviewed articles. The average number of pages was 14.32 pages per article. Figure 3 shows the distribution of DEA window analysis articles by number of pages. The minimum number of pages was four, while the maximum was 27, with a mode of 11 pages per article. About 27% of DEA window analysis articles were between 10 and 12 pages in length, while around 58% were between 8 and 15 pages in length. Finally, around 84% of articles were between 5 and 20 pages in length.

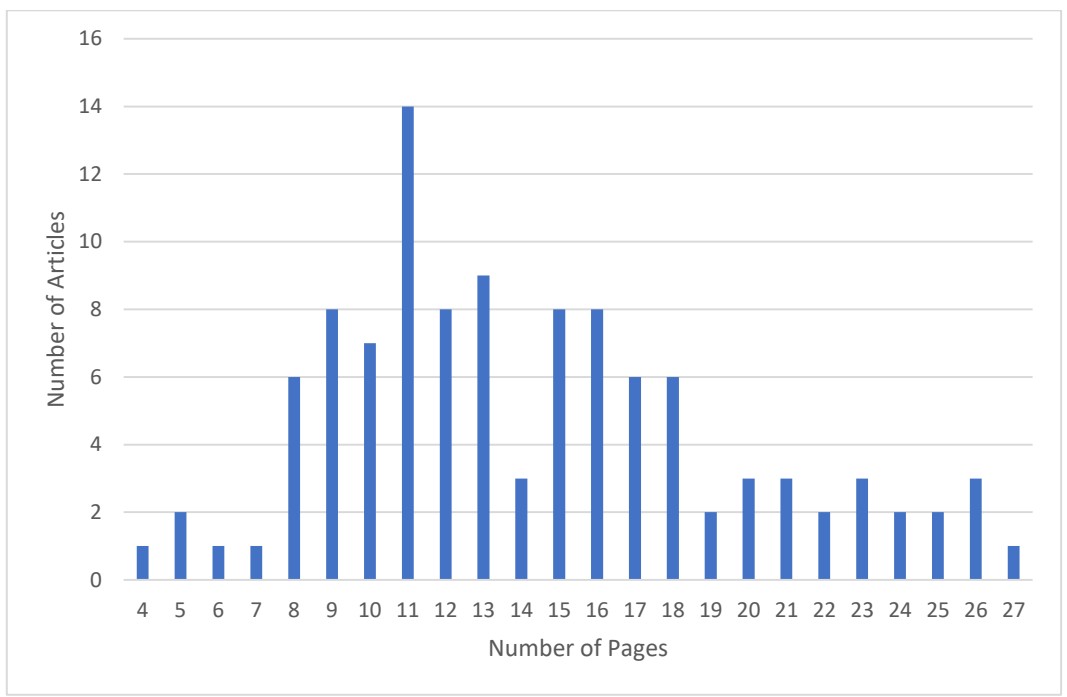

**Figure 3.** Distribution of DEA window analysis articles by the number of pages.

*5.3. Statistics Based on Number of Authors and Their Affiliations*

Figure 4 presents the number of authors per article, which ranged between 1 and 6 with an average of about 2.9 authors per article. Around 10% of articles were written by a single author and around 27% were written by two authors. The mode was three authors, representing 37% of articles published on DEA window analysis.

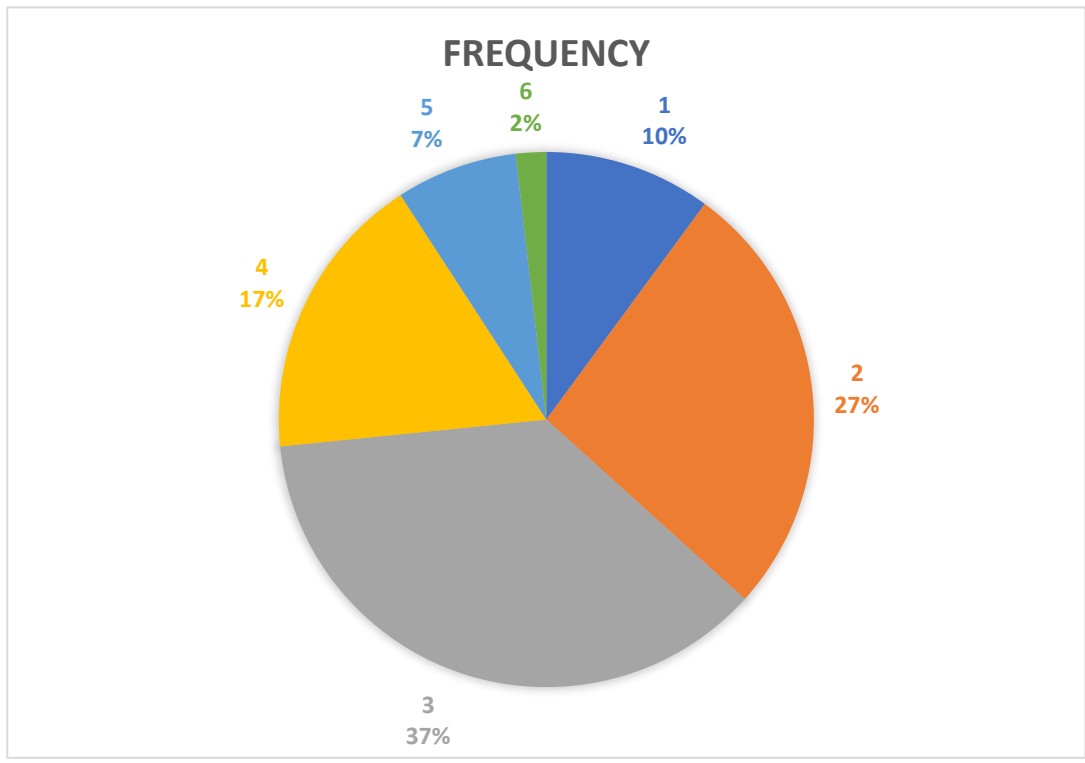

**Figure 4.** Frequency of authors per article.

Table 19 presents the affiliations of the first 19 institutions of the authors. The list contains only institutions that had five or more authors. The University of Thessaly was ranked first, with 17 authors contributing to the DEA window analysis literature. Islamic Azad University and the University of Science and Technology of China were both ranked second, as each had eight authors contributing to the DEA window analysis literature.

**Table 19.** List of the top 19 institutions that authors were affiliated with.

| No. | Institution | Frequency |
|---|---|---|
| 1 | University of Thessaly | 17 |
| 2 | Islamic Azad University | 8 |
| 2 | University of Science and Technology of China | 8 |
| 4 | Gazi University | 7 |
| 5 | Hunan University | 7 |
| 6 | Juraj Dobrila University of Pula | 7 |
| 7 | National Chiao Tung University | 6 |
| 8 | Shandong University | 6 |
| 9 | University of Jordan | 6 |
| 10 | University of São Paulo | 6 |
| 11 | University State of São Paulo | 6 |
| 12 | Wuhan University | 6 |
| 13 | Center for Energy and Environmental Policy Research | 5 |
| 14 | Hefei University of Technology | 5 |
| 15 | Technical University of Košice | 5 |
| 16 | University of Alcalá | 5 |
| 17 | University of Belgrade | 5 |
| 18 | University of Catania | 5 |
| 19 | University of Zagreb | 5 |

Similarly, Table 20 presents a list of the first 17 affiliated countries. Again, only countries that were affiliated with five or more authors are included. China was ranked number 1, with 101 authors contributing to the DEA window analysis literature, while Taiwan was ranked number 2.

**Table 20.** List of the top 17 countries that authors are affiliated with.

| No. | Country | Frequency |
|---|---|---|
| 1 | China | 101 |
| 2 | Taiwan | 37 |
| 3 | Greece | 23 |
| 4 | Brazil | 21 |
| 5 | USA | 20 |
| 6 | Spain | 19 |
| 7 | Croatia | 16 |
| 8 | Italy | 15 |
| 9 | Korea | 10 |
| 10 | Iran | 9 |
| 10 | Turkey | 9 |
| 12 | Australia | 8 |
| 13 | UK | 7 |
| 14 | Canada | 6 |
| 14 | Jordan | 6 |
| 14 | Slovakia | 6 |
| 17 | Serbia | 5 |

## 6. Conclusions

After reviewing the applications of DEA window analysis by analyzing 109 articles retrieved from the WoS database during the period 1996–2019, the number of articles published was found to be relatively small in the initial years but started growing in 2008

and reached a maximum of 27 articles in 2018. The articles were published in 79 distinct journals, with seven of them published in the journal *Sustainability*, followed by the journals *Applied Economics, Energy Policy, Expert System with Applications,* and *Journal of Cleaner Production*, each with four articles published. Moreover, the papers were classified into 15 distinct application areas. A total of 26 articles was classified into the energy and environment area, which had the highest number of published articles. This was followed by transportation, in which 12 articles were published. Furthermore, keyword terms were analyzed. The keyword term "DEA window analysis" and its variants appeared 69 times. This was followed by the keyword term "DEA" and its variants, which appeared 63 times. Additionally, the statistics of the lengths of the papers showed that the paper size ranged from 4 to 27 pages, with an average of 14.32 pages per article. Moreover, the number of authors ranged from a single author to six authors, with an average of 2.9 authors per article. Finally, the top institutions and countries the authors were affiliated with were tabulated. The University of Thessaly was ranked first among institutions, with 17 authors publishing articles in the field of DEA window analysis. Moreover, China was ranked first among countries, with 101 authors contributing to the DEA window analysis literature.

One limitation of this research is that it reviewed articles found only in the WoS database. The rationale behind this selection was to ensure that high quality journals be considered in this review, especially given that this is the first article reviewing DEA window analysis applications. To verify the results of this review and to gain a more comprehensive view, future research may review articles published in other databases such as Scopus and Google Scholar. One finding of this review is that there is potential to use DEA window analysis to evaluate the performance of companies in areas that have not been investigated, such insurance, construction, retailing, software, mining, etc.

One potential emerging related research area is optimization under uncertainties, where researchers are developing stochastic robust optimization models. For example, Qu et al. (2022) [121] measured the operational efficiency under uncertainties of an endowment insurance system in China using the robust DEA model. Qu et al. (2021) [122] included uncertainty cost into the maximum expert consensus model. Ji et al. (2022) [123] also considered uncertain parameters in their minimum cost consensus model. Thus, a future literature survey may investigate the robust optimization applications.

**Author Contributions:** Conceptualization, M.A.A.; methodology, M.A.A.; validation, T.A. and M.A.A.; formal analysis, M.A.A. and A.H.A.; resources, M.A.A. and A.H.A.; writing—original draft preparation, M.A.A.; writing—review and editing, T.A. and A.H.A.; supervision, T.A. All authors have read and agreed to the published version of the manuscript.

**Funding:** This research received no external funding.

**Institutional Review Board Statement:** Not applicable.

**Informed Consent Statement:** Not applicable.

**Data Availability Statement:** Not applicable.

**Acknowledgments:** The authors would like to acknowledge King Fahd University of Petroleum & Minerals (KFUPM) for its support.

**Conflicts of Interest:** The authors declare no conflict of interest.

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
