# Peer review of "A Survey of DEA Window Analysis Applications"

_processes, doi:10.3390/pr10091836_

Round 1
Reviewer 1 Report
The main purpose of this review paper is well laid out in the introduction part, together with the research questions.
In the abstract, please shortly state the scientific contribution of the paper.
Why haven’t you surveyed the Scopus database as well? I believe this kind of review surveying both WoS and Scopus databases would contribute greatly to the literature. Therefore, I would suggest including the Scopus database as well. However, this limitation is addressed in the conclusion part. (this decision is left upon the Editor)
Please rephrase this sentence in the Introduction Section: „As DEA is a huge field,“ (What do you mean a huge field??)
However, in my opinion, sub-chapters 4.1.-4.15. are not relevant. You do not qualitatively analyse all the published papers, therefore this is not of much scientific value. For instance, I would suggest leaving the Tables 3 – 17 only (without the text in subsections 4.1.-4.15.). If this is a review paper, it is recommended to either focus on the quantitative data and statistics or to focus on a qualitative and in-depth analysis of these surveyed papers. This is the reason the paper is so lengthy.
All in all, good job. Well structured and solid. A very interesting subject and a DEA technique that needs to be promoted and introduced to the scientific community.
Author Response
Reviewer 1
Comments and Suggestions for Authors
The main purpose of this review paper is well laid out in the introduction part, together with the research questions.
In the abstract, please shortly state the scientific contribution of the paper.
Response: we agree that a short statement of the scientific contribution will help to strengthen the importace of this paper. The contribution is shown below
“ To the best of the authors knowledge, this is the first survey reviewing the literature of the DEA window analysis applications in the 15 areas mentioned in the paper. .“
Why haven’t you surveyed the Scopus database as well? I believe this kind of review surveying both WoS and Scopus databases would contribute greatly to the literature. Therefore, I would suggest including the Scopus database as well. However, this limitation is addressed in the conclusion part. (this decision is left upon the Editor)
Response: It is true that the articles reviewed are from WoS, but most of these articles are also found in Scopus. However, future reserach can replicate and extend this work by incoproating articles published in other databases such as Scopus.
Please rephrase this sentence in the Introduction Section: „As DEA is a huge field,“ (What do you mean a huge field??)
Response: Maybe the sentence does not refelct the meaning intended. Here is the updated version
'' The basic CCR model of DEA has been extended to several versions, DEA window analysis is one of these several versions.''
However, in my opinion, sub-chapters 4.1.-4.15. are not relevant. You do not qualitatively analyse all the published papers, therefore this is not of much scientific value. For instance, I would suggest leaving the Tables 3 – 17 only (without the text in subsections 4.1.-4.15.). If this is a review paper, it is recommended to either focus on the quantitative data and statistics or to focus on a qualitative and in-depth analysis of these surveyed papers. This is the reason the paper is so lengthy.
Response: We see the point of this comment. As stated, we can remove the subsections 4.1-4.15 and leave the tables 3-17. This will help to reduce the size of the paper and will provide more focus to the readers. In the future, an article may analyze these papers qualitiavely as suggested by the reviewer.
All in all, good job. Well structured and solid. A very interesting subject and a DEA technique that needs to be promoted and introduced to the scientific community.

Reviewer 2 Report
Please see the attachment.

Author Response
Reviewer 2
Title: A Survey of the DEA Window Analysis Applications
- Please briefly explain the reasons for dividing the reviewed articles into 15 application areas in this paper.
Response:
“DEA applications are classified into 26 fields (Liu, J.S.; Lu, L.Y.Y.; Lu, W.-M.; Lin, B.J. A survey of DEA applications. Omega 2013, 41, 893–902, doi:10.1016/j.omega.2012.11.004.). However, only 15 fields are selected in this paper due to the applications relevant to Window DEA.
- Some details regarding the formula in section 2 should be provided.
Response: The basic CCR model is stated then transformed into linear programming model. This is followed by introducing “windows” and defining input/output matrices. If there are additional details missing, we are glad to add them.
- The literature review in this paper lacks some outlook for the future and needs to further summarize the shortcomings and look forward to the future research trends of this study. There are many uncertainties for the study. For example, the robust DEA model is proposed in the uncertain environment by Qu et al (2022). The authors are recommended to discuss the results with the presents of uncertainties. The relevant publications for uncertainties can be cited: Qu, S.; Li, Y.; Ji, Y. The mixed integer robust maximum expert consensus models for large-scale GDM under uncertainty circumstances. Appl. Soft Comput. 2021, 107, 107369. Ji, Y.; Li, H.; Zhang, H. Risk-averse two-stage stochastic minimum cost consensus models with asymmetric adjustment cost. Group Decis. Negot. 2022, 31, 261–291.
Response: The main purpose of this study is to provide a literature review of the applications of the DEA Window analysis. One major benefit of DEA window analysis is to measure the efficiency of the DMUs over time and track the changes of such efficiencies. As mentioned in the first point, there are 26 possible application areas for DEA. This study identified only 15 areas. Clearly, there is high potential to use the DEA window analysis in other areas that can benefit from its advantage of measuring the efficiency over time. The following part is included in the conclusion:
“One potential emerging related research area is optimization under uncertainties where researchers are developing stochastic robust optimization models. For example, Qu et al. (2022) [121] measured the operational efficiency under uncertainties of endowment insurance system in China using robust DEA model. Qu et al. (2021) [122] took uncertainty cost into the maximum expert consensus model. Ji et al. (2022) [123] also considered uncertain parameters into their minimum cost consensus model. So future literature survey may investigate the robust optimization applications. “
- It is noted that your manuscript needs careful editing by someone with expertise in technical English editing paying particular attention to English grammar, spelling, and sentence structure so that the goals and results of the study are clear to the reader.
Response: the paper has been checked linguistically by the English Editing Services provided by MDPI.
- Rreference
Qu, S.; Feng, C.; Jiang, S.; Wei, J.; Xu, Y. Data-Driven Robust DEA Models for Measuring Operational Efficiency of Endowment Insurance System of Different Provinces in China. Sustainability 2022, 14, 9954. https://doi.org/10.3390/su14169954

Reviewer 3 Report
The work addresses the DEA Window Analysis Applications. However the need for this study is not clearly stated at the end of the Introduction. Why does the scientific community need this manuscript?
Added value of the paper: The review discusses several applications within the different sectors.
Figures and tables: Formatting of the figures and tables should be improved. All should be consistent.
Amendments:
Summary tables should be included in each subsection.
Is there a correlation among different areas? Is there a significant trend for publications in several areas at the same time? This should be also studied.
Author Response
Reviewer 3
The work addresses the DEA Window Analysis Applications. However the need for this study is not clearly stated at the end of the Introduction. Why does the scientific community need this manuscript?
Added value of the paper: The review discusses several applications within the different sectors.
Response: DEA window analysis is an extension of the basic CCR model that is used to measure the efficiency of the DMUs over time. As stated in the comment, this paper surveyed the application of DEA window analysis over 15 application areas. In fact, this approach can be used in other application areas such as insurance, automobile, fisheries, etc. By surveying the applications, researchers can use this technique to study the efficiency of DMUs in other sectors not searched yet. The following sentence is added at the end of the introduction.
“This work surveys the application of DEA window analysis over 15 sectors. To the best of the authors knowledge, this is the first time such a survey takes place which is expected to be appreciated by the scientific community.”
Figures and tables: Formatting of the figures and tables should be improved. All should be consistent.
Response: It true that the format of the figures and tables should be improved. We think these tables and figures will be better formatted during the proofreading stage.
Amendments:
Summary tables should be included in each subsection.
Response: One reviewer suggested to remove the subsections 4.1-4.15 and keep the tables. We removed the subsections, and the manuscript is more concise.
Is there a correlation among different areas? Is there a significant trend for publications in several areas at the same time? This should be also studied.
Response: The following table shows the number of publications across the 15 application areas over the period 1996-2019. Over the period 1996-2007, there are only 0 or 1 publication in total for all 15 areas. The total number of publications increased to be between 2 to 7 over the period 2008-2015. This number increased in the period 2016-2019 to be between 12 and 27. However, the number of publications in each area in a specific year ranges between 0 and 4 except for energy & environment in 2018 in which there are 9 publications. Given these number, it is difficult to find reasonable correlation among the different areas. One obvious observation is there is an increase in the use of the DEA window analysis in recent years. This justifies the importance of conducting such a review paper to help researchers who are interested to use this technique to measure the efficiency of the DMUs in other application areas.

Round 2
Reviewer 2 Report
The authors have cleared all of my concerns. So I think this paper can be accepted now.